# Electrical coupling controls dimensionality and chaotic firing of inferior olive neurons

**Huu Hoang**[1]*, **Eric J. Lang**[2], **Yoshito Hirata**[3,4,5], **Isao T. Tokuda**[6], **Kazuyuki Aihara**[3,5], **Keisuke Toyama**[1], **Mitsuo Kawato**[1,7]*, **Nicolas Schweighofer**[8]*

**1** Computational Neuroscience Laboratories, ATR Institute International, Kyoto, Japan, **2** Department of Neuroscience and Physiology, New York University School of Medicine, New York, New York, United States of America, **3** Institute of Industrial Science, The University of Tokyo, Tokyo, Japan, **4** Mathematics and Informatics Center, The University of Tokyo, Tokyo, Japan, **5** International Research Center for Neurointelligence (WPI-IRCN), The University of Tokyo, Tokyo, Japan, **6** Department of Mechanical Engineering, Ritsumeikan University, Shiga, Japan, **7** RIKEN Center for Advanced Intelligence Project, ATR Institute International, Kyoto, Japan, **8** Biokinesiology and Physical Therapy, University of Southern California, Los Angeles, California, United States of America

* hoang@atr.jp (HH); kawato@atr.jp (MK); schweigh@pt.usc.edu (NS)

**Data Availability Statement:** All relevant data are within the manuscript and its Supporting Information files.

## Abstract

We previously proposed, on theoretical grounds, that the cerebellum must regulate the dimensionality of its neuronal activity during motor learning and control to cope with the low firing frequency of inferior olive neurons, which form one of two major inputs to the cerebellar cortex. Such dimensionality regulation is possible via modulation of electrical coupling through the gap junctions between inferior olive neurons by inhibitory GABAergic synapses. In addition, we previously showed in simulations that intermediate coupling strengths induce chaotic firing of inferior olive neurons and increase their information carrying capacity. However, there is no *in vivo* experimental data supporting these two theoretical predictions. Here, we computed the levels of synchrony, dimensionality, and chaos of the inferior olive code by analyzing *in vivo* recordings of Purkinje cell complex spike activity in three different coupling conditions: carbenoxolone (gap junctions blocker), control, and picrotoxin (GABA-A receptor antagonist). To examine the effect of electrical coupling on dimensionality and chaotic dynamics, we first determined the physiological range of effective coupling strengths between inferior olive neurons in the three conditions using a combination of a biophysical network model of the inferior olive and a novel Bayesian model averaging approach. We found that effective coupling co-varied with synchrony and was inversely related to the dimensionality of inferior olive firing dynamics, as measured via a principal component analysis of the spike trains in each condition. Furthermore, for both the model and the data, we found an inverted U-shaped relationship between coupling strengths and complexity entropy, a measure of chaos for spiking neural data. These results are consistent with our hypothesis according to which electrical coupling regulates the dimensionality and the complexity in the inferior olive neurons in order to optimize both motor learning and control of high dimensional motor systems by the cerebellum.

**Funding:** HH, KT and MK were supported by a contract with the National Institute of Information and Communications Technology entitled 'Development of network dynamics modeling methods for human brain data simulation systems' (Grant No.173) and 'Analysis of multi-modal brain measurement data and development of its application for BMI open innovation' (Grant No.209). HH and KT are partially supported by ImPACT Program of Council of Science, Technology and Innovation (Cabinet Office, Government of Japan). HH, ITT and KT are partially supported by MEXT Kakenhi (No. 17H06313). HH and KT are partially supported by JST ERATO (JPMJER1801, "Brain-AI hybrid"). KA is partially supported by MEXT Kakenhi (No. 15H05707) and AMED under Grant Number JP20dm0307009. EJL acknowledges grants NIH NS095089 and NS37028. NS acknowledges grants NSF BCS-1031899 and 1R56NS100528-01. The funders had no role in study design, data collection and analysis, decision to publish, or preparation of the manuscript.

**Competing interests:** The authors have declared that no competing interests exist.

## Author summary

Computational theory suggests that the cerebellum must decrease the dimensionality of its neuronal activity to learn and control high dimensional motor systems effectively, while being constrained by the low firing frequency of inferior olive neurons, one of the two major source of input signals to the cerebellum. We previously proposed that the cerebellum adaptively controls the dimensionality of inferior olive firing by adjusting the level of synchrony and that such control is made possible by modulating the electrical coupling strength between inferior olive neurons. Here, we developed a novel method that uses a biophysical model of the inferior olive to accurately estimate the effective coupling strengths between inferior olive neurons from *in vivo* recordings of spike activity in three different coupling conditions. We found that high coupling strengths induce synchronous firing and decrease the dimensionality of inferior olive firing dynamics. In contrast, intermediate coupling strengths lead to chaotic firing and increase the dimensionality of the firing dynamics. Thus, electrical coupling is a feasible mechanism to control dimensionality and chaotic firing of inferior olive neurons. In sum, our results provide insights into possible mechanisms underlying cerebellar function and, in general, a biologically plausible framework to control the dimensionality of neural coding.

## Introduction

The cerebellum plays important roles in motor learning and motor control, although how it performs these roles is still unclear. In particular, the role of the inferior olive (IO) continues to be debated. On one hand, there is evidence that the olivo-cerebellar system conveys error signals into the cerebellum [1–4] and induces plasticity in parallel fiber-Purkinje cell synapses (e.g., [5–8]). Such error-driven plasticity is a central tenet of the original motor learning theory of the cerebellum [8–10], as it can allow learning of internal models for motor control [11–16]. On the other hand, there is also evidence that olivo-cerebellar activity has a direct role in generating ongoing motor commands because of its ability to dynamically generate large ensembles of synchronously active Purkinje cells during movement that can affect downstream motor systems [17–20]. Moreover, it has been shown that spontaneous olivo-cerebellar activity can directly influence ongoing spiking in cerebellar nuclear cells, which relay motor commands produced by the cerebellar cortex [21,22].

However, a fundamental and outstanding question that needs to be addressed by both theories is: how does the olivo-cerebellar system convey information, whether for learning or for controlling the high dimensional and nonlinear motor systems that generate movements, despite the low-firing rates of inferior olive neurons (typically ~1 Hz). Indeed, olivary neurons discharge at most one or two times during a typical movement [23]. For motor learning, compared to an artificial learning machine that can use high frequency errors [24], such a low firing rate significantly decreases the error transmission capability of the system, and thus its learning efficiency. Similarly, for motor control, the low firing rate presents a problem for the direct participation of the olivo-cerebellar system in the generation of high frequency and high dimensional motor commands across multiple muscles and joints.

An answer to this question is suggested by the unusual organization of the IO, in which neurons form the strongest electrically coupled neuronal network in the adult mammalian brain [25–29]. This coupling underlines synchronization of complex spike activity in Purkinje cells [30–33]. Moreover, the patterns and extent of synchronization are dynamically controlled by two types of synaptic inputs to the IO: GABAergic synapses whose activity reduces

synchrony [33,34] and excitatory synapses whose activity alters the distribution of synchrony and enhances IO coupling of weakly coupled neurons [35–37].

We have previously proposed that the capacity of the olivo-cerebellar system to adaptively control the dimensionality of the IO firing dynamics, defined as the minimal dimension required to provide a precise description of the neural dynamics, via modulation of electrical synapses between IO neurons is central to answering the above question [38–40]. According to this idea, when coupling is high, IO synchrony is high, and groups of related neurons in the olivo-cerebellar system behave, in the limit, as a single-neuron chain, decreasing the dimensionality of the IO firing dynamics to one. For motor control, synchronous IO signals would induce synchronous activation of Purkinje cell ensembles, which, in turn, would tune the downstream systems to facilitate the initiation and coordination of fast and crude movements [19]. For motor learning, high synchrony in the early stages of learning induces strong plasticity at the parallel-fiber-Purkinje-cell synapses of large numbers of Purkinje cells simultaneously, resulting in fast but crude learning [39,40]. In contrast, *in silico* computer simulations show that chaotic resonance occurs when coupling is decreased, leading to a decrease in synchrony [41–45]. Chaotic resonance can thereby allow both sophisticated learning and control, either for the final subtle corrections to optimize movements or for control of fine movements [45,46].

Here, we re-analyzed *in vivo* recordings of complex spikes recorded simultaneously from arrays of Purkinje cells [30,33,34] under three pharmacologically induced coupling conditions (low, control, high) to study the effect of coupling on the dimensionality of the IO code and on the induction of chaotic resonance. The low coupling condition was generated by intra-IO injection of the gap junction blocker carbenoxolone (CBX), which lowers complex spike synchrony [30], whereas the presumed high coupling condition was generated by intra-IO injection of the GABA-A blocker picrotoxin (PIX), which increases complex spike synchrony [33,34]. In the present study, we tested our two predictions that 1) increasing the synchrony level, via increased electrical coupling between IO neurons, decreases the dimensionality of IO firing dynamics and 2) intermediate coupling induces chaotic spiking.

## Results

### Estimation of the effective coupling between IO neurons *in vivo*

To examine the effect of electrical coupling on the dimensionality and chaotic dynamics of the IO code, we first need to determine the physiological range of effective coupling strengths between IO neurons under *in vivo* conditions. Direct quantitative measurement of electrical coupling between IO neurons has been obtained in slice preparations [47–49]; however, such measurements remain technically impossible *in vivo*. We therefore used an indirect approach. Purkinje cell complex spikes, as opposed to simple spikes, bear a one-to-one relationship to IO discharges. Thus, complex spikes can be used as a proxy for IO spikes (see S1 Fig for examples of complex spike recordings in these three conditions; see Methods for experimental procedures). Note that an IO cell discharge can lead to one or several axonic spikes occurring with inter-spike intervals on the order of a millisecond [50]. Each such discharge, whether composed of one or several spikes, leads to a single complex spike in the Purkinje cells to which the IO cell projects. In this paper we consider each such IO discharge as a single 'spike' event. *In vivo* complex spike activity was compared with simulated activity generated by a biophysical model of a network of coupled IO neurons, whose parameters were estimated via a Bayesian method [51] that we modified by using Bayesian model-averaging to improve the robustness of its estimation of the coupling parameters. The coupling parameters that produced the

spatiotemporal firing patterns that best matched those of the experimental data were used as the estimates (see below for details).

The biophysical IO model was adapted from a model that we previously developed to investigate the effect of PIX in modifying coupling via its action of blocking GABA inhibitory synapses. This model modified the original IO model [43,52] to include the modulation of electrical conductance between IO cells via inhibitory inputs from deep cerebellar nuclear cells (for review of IO anatomy and function, see [53]). Briefly, in the model, each IO neuron comprises a soma, a main dendrite, and four dendritic spine compartments, with each compartment having distinct ionic conductances. Most notably, the dendritic compartment has a high threshold calcium conductance and a calcium-activated potassium conductance, which are responsible for the after-depolarization and after-hyperpolarization sequence that follows each sodium spike and for the low firing rates of IO neurons [52,54,55]. Each neuron is coupled to its neighbors via electrical coupling conductances between the spine compartments. An inhibitory synaptic conductance in the spine compartment modulates the effective coupling strength (for a description of the model, see Methods for details and [51,56]; the code of the model is available for download, see Code Availability). In the present study, we increased chaotic dynamics by increasing the sodium conductance, as tests showed that these changes better accounted for the actual IO firing properties (see Methods for details).

Using the model, it is possible to derive a theoretical "effective" electrical coupling conductance $g_{eff}$ as a function of the axial conductance of the spines $g_s$, the electrical coupling conductance $g_c$, and the GABAergic synaptic conductance $g_i$ (see [57] and Methods for details). Estimates of $g_c$ and $g_i$, were obtained by comparing sixty-seven spatiotemporal features–including firing rates, local variation [58], minimal distance [57], auto-correlograms and cross-correlograms–of the model's spike activity to those of the complex spike data sets for different values of synaptic noise input frequencies ($g_s$ was held constant, see Methods for details). A final estimate of $g_i$ and $g_c$ for each condition (CBX, CON, and PIX) was obtained using Bayesian averaging of the model estimates for the different synaptic noise levels, weighted in proportion to the goodness-of-fit of the model for each noise level (see S3 Fig and S4 Fig, and Methods for details). Note that for this analysis, we used data from *in vivo* neurons whose activity was clearly affected by the drug treatments, based on changes in their firing rate from control levels (see S2 Fig and Methods for details). This was done because the lack of effect in some cells likely reflects the experimental limitation that the drug injection was localized to one part of the IO, whereas climbing fibers to the recording array arise from multiple IO regions (see [30] for results and discussion of this issue).

In order to compare the coupling across all three conditions, the two CON groups (CON–CBX and CON–PIX) were combined. As expected, the estimated inhibitory conductance $g_i$ in the CON condition (Fig 1A– 1.15 ± 0.21 mS/cm$^2$, $n$ = 100 neurons; all results reported as *mean ± std*) was significantly higher than in the PIX condition (0.72 ± 0.3 mS/cm$^2$, $n$ = 47 neurons; PIX vs CON: p < 0.0001). However, $g_i$ in the CBX condition (1.02 ± 0.13 mS/cm$^2$, $n$ = 53 neurons) was also significantly smaller than in both its own CON condition (1.21 ± 0.21 mS/cm$^2$, CBX vs CON–CBX, p < 0.001) and the combined CON condition (CBX vs CON: p < 0.001) probably due to possible effects of CBX on inhibitory synapses [59]. Similarly, the estimated gap-junctional conductance $g_c$ in the CBX condition (Fig 1B– 0.88 ± 0.22 mS/cm$^2$) was significantly smaller than in the CON condition (1.19 ± 0.25 mS/cm$^2$, CBX vs CON: p < 0.0001), but there was no significant difference between the PIX (1.16 ± 0.21 mS/cm$^2$) and CON conditions (PIX vs PIX–CON, p = 0.2; PIX vs CON: p = 0.5). As a result of these changes in $g_i$ and $g_c$, the estimated effective coupling strength, $g_{eff}$, differed across the three conditions (one-way ANOVA: p < 0.0001). $g_{eff}$ was smallest for the CBX condition (Fig 1C–$g_{eff}$ = 0.030 ± 0.002 mS/cm$^2$, CBX–CON: p < 0.0001), intermediate for the CON condition ($g_{eff}$ =

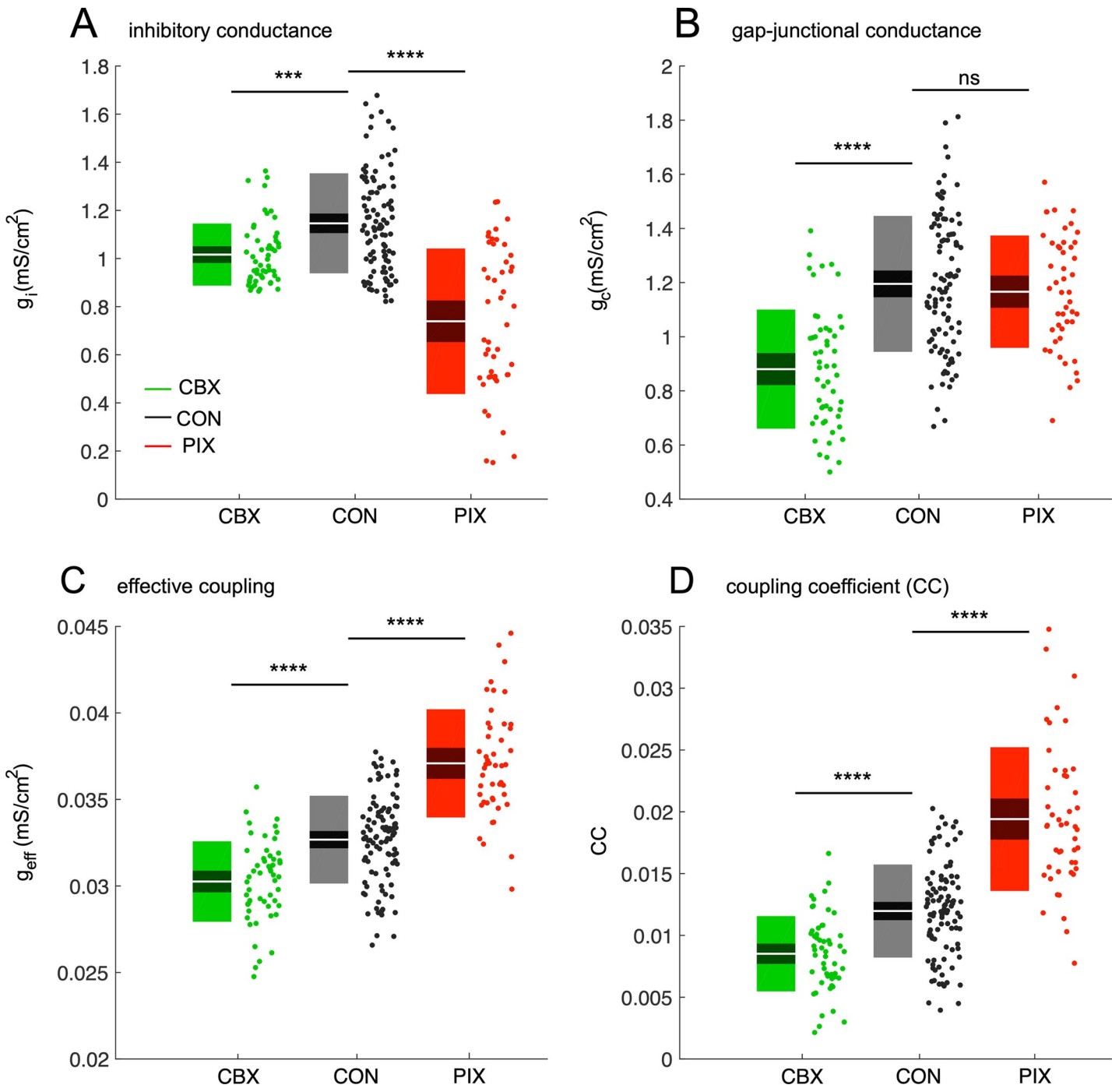

**Fig 1. Estimation of conductance and coupling coefficients in the three experimental conditions.** A-B: Values of the GABAergic synaptic conductance $g_i$ (A) and electrical coupling conductance $g_c$ (B) estimates for the three experimental conditions: carbenoxolone (CBX, green box), control (CON, black box) and picrotoxin (PIX, red box). C: The effective coupling coefficient $g_{eff}$ computed from Eq (1) for the three conditions. D: The coupling coefficient (CC) estimated via simulations for the three conditions. Each boxplot shows white line as the mean, dark region as 95% CIs and light region as 1 std. Asterisks represent significance levels: ns p > 0.05, ***p < 0.001, ****p < 0.0001.

$0.032 \pm 0.002$ mS/cm$^2$) and largest for the PIX condition ($g_{eff} = 0.036 \pm 0.003$ mS/cm$^2$, PIX–CON: p < 0.0001).

We confirmed these results without pooling the two CON groups. The best fit of the model to the data indicated PIX and CBX reduced mean $g_i$ and $g_c$ approximately 30% and 25% from their respective CON values, consistent with their known pharmacological effects. Specifically, $g_i$ was significantly decreased in the PIX condition from its CON value ($1.06 \pm 0.18$ mS/cm$^2$, CON–PIX vs PIX, $p < 0.0001$). Similarly, $g_c$ in the CBX condition was significantly smaller than in its CON condition ($1.18 \pm 0.28$ mS/cm$^2$, CON–CBX vs CBX, $p < 0.0001$).

Next, we examined whether the estimates of effective coupling strength, $g_{eff}$, were biologically realistic by computing the coupling coefficients (CCs) as the average ratio of the change in steady state membrane potentials of a master cell and its four neighboring cells in response to a current step (see S5 Fig and Methods for details). The calculated CCs for our data were similar to *in vitro* values [49]. As expected, CC was smaller in the CBX condition (Fig 1D–$CC = 0.008 \pm 0.002$, CBX vs CON: $p < 0.0001$) and larger in the PIX condition ($CC = 0.019 \pm 0.006$, PIX vs CON: $p < 0.0001$) than in the CON condition ($CC = 0.012 \pm 0.003$).

The estimated $g_i$ and $g_c$ parameters were then used to generate simulated spike trains under all three conditions. In each case, the spike trains were comparable to those of the recorded complex-spike activity (see Fig 2A–2C). Quantitatively, firing rates (model: $0.39 \pm 0.30$ and $1.43 \pm 0.76$ and $2.21 \pm 0.53$; data: $0.42 \pm 0.27$ and $1.34 \pm 0.72$ and $2.83 \pm 1.20$ for CBX, CON and PIX conditions, respectively) and cross-correlations (model: $0.01 \pm 0.01$ and $0.04 \pm 0.03$ and $0.11 \pm 0.04$; data: $0.02 \pm 0.01$ and $0.06 \pm 0.02$ and $0.18 \pm 0.05$; 10 ms time bin) increased in the PIX and decreased in the CBX condition. In contrast, auto-correlations (model: $0.69 \pm 0.20$ and $0.48 \pm 0.17$ and $0.35 \pm 0.09$; data: $0.75 \pm 0.13$ and $0.51 \pm 0.17$ and $0.32 \pm 0.12$; 50 ms time bin) showed the opposite change, being lower in the PIX and higher in the CBX condition (Fig 2D and 2E). The changes in those spike train measures reflect the changes in the firing dynamics, which became more synchronous across the IO neuronal ensemble under the PIX than the CON condition, and became less so under the CBX condition. The strong agreement of these measures between the experimental and model data confirms the goodness-of-fit of the model in all three data conditions.

## Dimensionality is inversely related to synchrony and effective coupling levels

Next, we examine the relationship between dimensionality and synchrony in an identical time bin of those two measures. The dimensionality $d$ is defined as the minimal number of principal components accounting for approximately 90% of the variability in the covariance data (see Eq 5 in Methods), as proposed by [60]. We extracted the average firing rates of neurons in 50-second long periods and applied principal component analysis (PCA) to compute the covariance of these firing rate vectors for each animal. $d$ values were then normalized by the number of neurons simultaneously recorded (see S6 Fig and Methods for details). Synchrony was measured by calculating the zero-lag cross-correlation coefficient of the two spike trains (Eq 4). In the literature, time bins of 1–10 ms have been commonly used [22,61–63] (see [22] for a detailed justification of this range from the perspective of the impact on cerebellar nuclear cells). Thus, in our study, we chose a time bin of 10 ms for computing both the synchrony and the dimensionality of Purkinje complex spike activity (see Methods for details).

For the present dataset, and consistent with the parent datasets [30,33,34], synchrony levels increased three-fold in the PIX condition (synchrony = $0.186 \pm 0.05$, PIX vs CON, $p = 0.012$) and decreased about three-fold in the CBX condition (synchrony = $0.018 \pm 0.009$, CBX vs CON, $p = 0.0015$) compared to the CON condition (synchrony = $0.061 \pm 0.015$)–Fig 3A. We then assessed whether the dimensionality changed with the drug condition. As predicted, $d$ differed across the three conditions (one-way ANOVA, $p = 0.02$), being smaller in the PIX

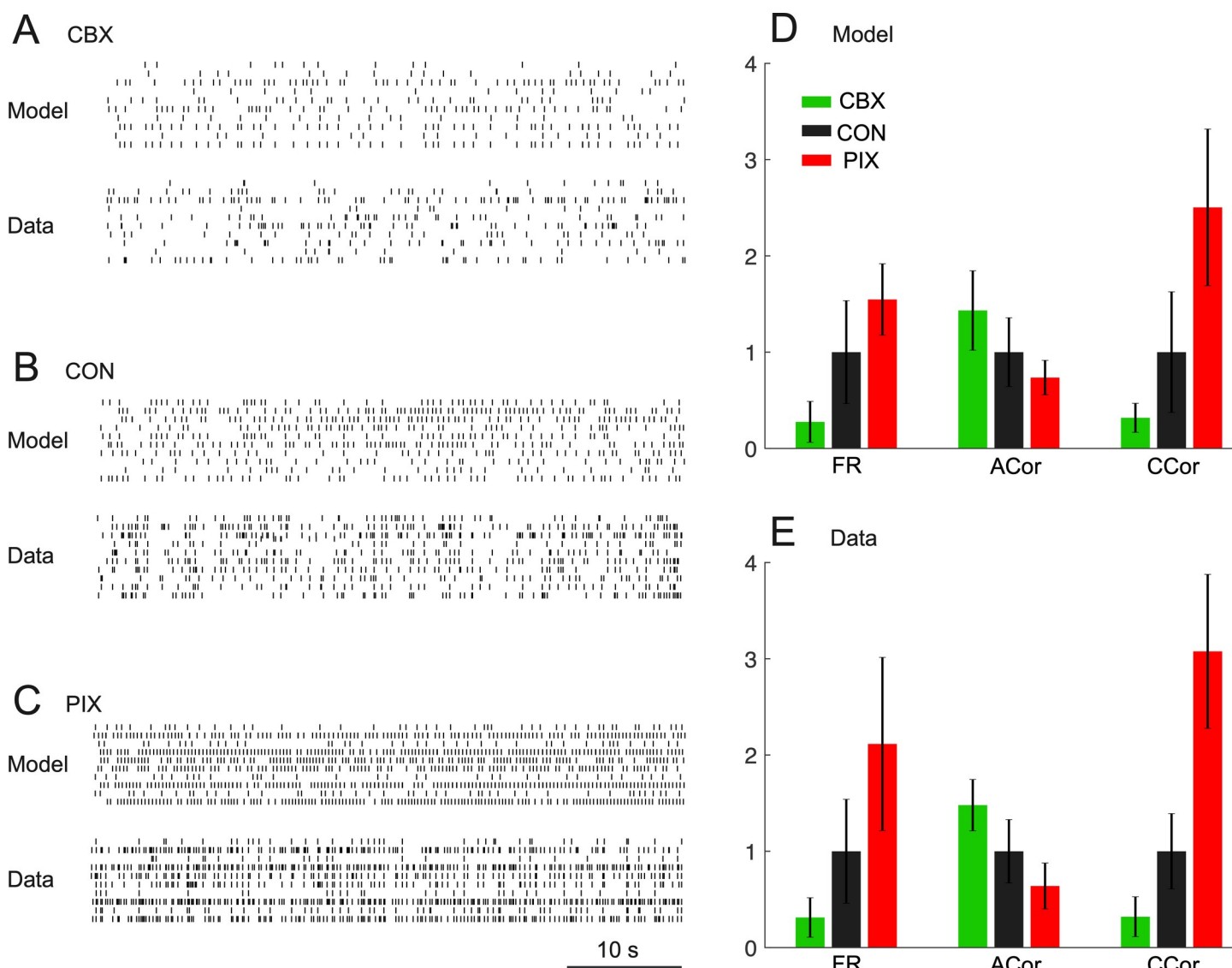

**Fig 2. Similarity between IO firing for model and data.** A–C: Raster plots of ten representative IO neurons of the model and the experimental complex spike data of three animals in the three conditions. Each row of tick marks represents the activity of a single neuron. A. Carbenoxolone (animal #1, irregular spiking). B. Control (animal #7, oscillatory spiking) C. Picrotoxin (animal #9, highly synchronous spiking). D–E: Three major spatiotemporal features extracted from the spike trains–firing rates, auto-correlations and cross-correlations–in the three data conditions of the model (D) and the data (E). FR: firing rates. ACor: auto-correlation. CCor: cross-correlation. The ordinates of D–E are scaled so that the mean value of the CON is 1.

condition ($d = 0.15 \pm 06$, PIX vs CON, p = 0.07) and larger in the CBX condition ($d = 0.40 \pm 0.16$, CBX vs CON, p = 0.07) than in the CON condition ($d = 0.25 \pm 0.11$)–Fig 3B. Although change in the dimensionality was large between conditions (approximately 2-fold from CBX to CON and from CON to PIX), there was no statistical difference between both the PIX and CBX conditions with the control group. This is probably due to the small number of samples as well as the large variance in the dimensionality across the animals in each condition.

The relationship between changes in synchrony and dimensionality was consistent and was seen in each animal (Fig 3C). Relative to CON, a 2-fold decrease of synchrony in the CBX condition was associated with an ~70% increase in *d*, whereas a 3-fold increase of synchrony in the PIX condition was associated with a 40% reduction in dimensionality. Furthermore, we

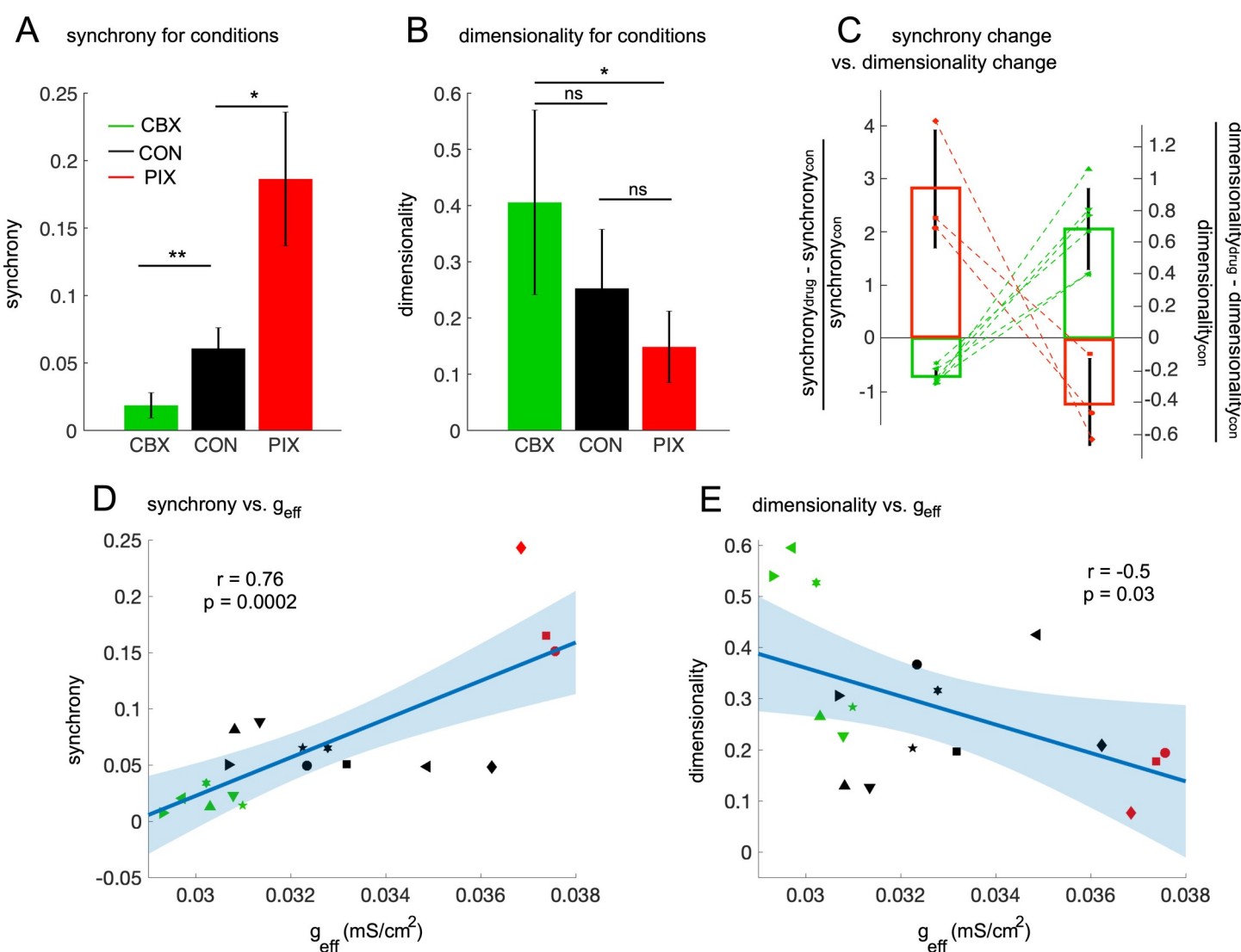

**Fig 3. The synchrony and the dimensionality in IO firings moderated by effective coupling.** A-B: The synchrony (A) and the dimensionality (B) for each of the three data conditions. Significance level, ns p > 0.05, * p < 0.05, ** p < 0.01. C: the change of synchrony by drugs (either CBX–green bars, or PIX–red bars, left axis) is coupled with change in the dimensionality (right axis) compared with the control level. Each connected pair of data points is from one animal. D–E: The synchrony (D) and the dimensionality (E) as functions of effective coupling strength averaged for selected neurons in individual animals confirming that effective coupling is a control parameter to optimize the synchrony and thus the dimensionality of IO firings. Each type of symbol represents the data of an individual animal. The cyan solid lines show results of the linear regression models and shaded regions are of 95% CIs.

divided the control periods of the complex spike data (n = 9 animals) into short time segments, to assess the relationship between synchrony level and dimensionality for levels that are within the same range as occurs in the awake animal [63]. Synchrony levels fluctuated between these segments and these fluctuations were negatively correlated with changes in the dimensionality (S7 Fig). Thus, variations in synchrony within physiological range are associated with significant changes in dimensionality.

Next, we quantitatively investigated the effect of effective coupling on synchrony and dimensionality. As expected, there was a positive correlation (r = 0.76; p = 0.0002, Fig 3D) between synchrony and effective coupling averaged for each animal. Note that we found an outlier (animal #9 in the PIX condition, the red diamond in Fig 3D) by computing Cook's

distances with a threshold of five times the mean value [64]. But even when that outlier was removed, the correlation was still significant (r = 0.75; p = 0.0004). In addition, there was a negative correlation between $g_{eff}$ and $d$ (r = -0.5; p = 0.03, no outlier detected with the same criteria above, Fig 3E). These results support our hypothesis that synchronization is a feasible mechanism for dimensionality reduction in IO neurons and that effective coupling is a control parameter that the IO uses to optimize the dimensionality of the olivo-cerebellar system.

### Inverted U-shaped relationship between complexity entropy and effective coupling

We next addressed the question of whether intermediate, physiological coupling strengths maximize the chaotic level of IO activity. Lyapunov exponents quantify the sensitivity of a dynamical system to the initial conditions [65,66], and are thus often used as indicators of chaos. However, methods to compute Lyapunov exponents from time series data [67,68] are not applicable to our spike data sets, because the computation requires access to continuous variables. We therefore computed the complexity entropy, which is applicable to spike train data and approximates the largest Lyapunov exponents in simulations of IO neurons [69,70] (see S8 Fig and Methods for details).

For both the simulated IO spike and the experimental complex spike data sets, we investigated whether the relationship between complexity entropy and effective coupling formed an inverted U-shape, as previously shown in simulations [43,45]. For each of the experimental IO neurons, we computed the complexity entropy from the simulated spike data that was generated with the estimated coupling values that best fit the data in terms of the PCA error (difference between experimental and simulated spike data in the PCA space, S4A Fig). For the IO model (Fig 4A), the second order model (regression model in Wilkinson notation [71]: *entropy ~ 1 + $g_{eff}$ + $g_{eff}^2$*, Bayesian information criterion (BIC): -1263.4) where *entropy* is the complexity entropy, had a negative coefficient for the second order term (*mean ± sem*, -157 ± 36), and better fit the simulated spikes in the three conditions than the first-order linear model (*entropy ~ 1 + $g_{eff}$*, BIC: -646.8; Log likelihood ratio test (LLR): p < 0.0001). For the IO data (Fig 4B), a mixed effect regression model analysis, with *Animal* as the random intercept accounting for repeated measures within the same animal, showed that the second order model (*entropy ~ 1 + $g_{eff}$ + $g_{eff}^2$ + (1 | Animal)*, BIC: -1319.5*)*, where *(1 | Animal)* is the random intercept, had a negative fixed-effect coefficient of the second order term (*mean ± sem*, -75 ± 30), and provided a better fit than the linear model (*entropy ~ 1 + $g_{eff}$ + (1 | Animal)*, BIC: -1318; LLR: p = 0.01). We further conducted a Gaussian-Process regression, which does not assume an explicit relationship between the coupling and the complexity entropy. The result also showed an inverted U-curve that peaks at around $g_{eff}$ = 0.033 mS/cm$^2$ for both the model and the data (S9 Fig). These results all indicate that intermediate coupling strengths induce chaotic behavior in both the model and the data. Note that the relatively small changes in the complexity entropy that we observed in the model and data correspond to large changes in the largest Lyapunov exponent $\lambda_1$ (see Methods and S8B Fig), from synchronous and rhythmic firings (in both the model and the data, *entropy* = 0.21, $\lambda_1$ = 5 bits/second) to chaotic firings (*entropy* = 0.24, $\lambda_1$ = 40 bits/second).

## Discussion

### Estimation of the physiological electrical coupling between IO neurons from complex spike data

We developed a novel technique that combines computational modeling, Bayesian inference and model-averaging to estimate the effective coupling among IO neurons from rat *in vivo*

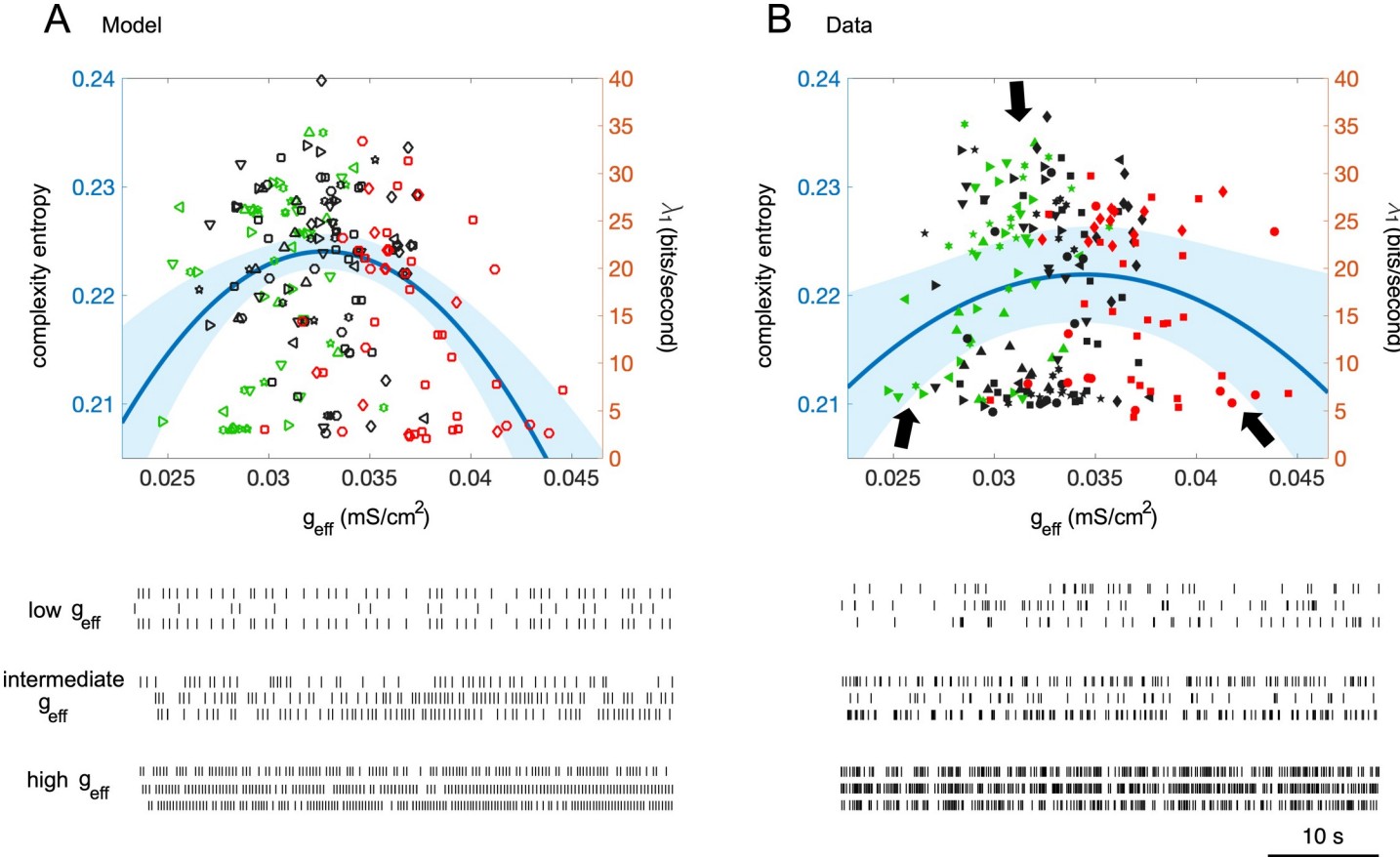

**Fig 4. Inverted U-shaped relationship of complexity versus effective coupling, model and data.** A-B: Complexity entropy versus effective coupling. Upper panel: chaotic levels measured by the complexity entropy of the spike data as a function of effective coupling strength for the model (A) and real IO neurons (B) confirming that moderate couplings induce chaos. Each value in (A) (open symbols) is given by the model neuron that best fits to the actual IO neuron in terms of the PCA error. The right ordinates of A-B represent the first Lyapunov exponents approximated from the simulation data (S8B Fig). Each type of symbol in (B) represents the data of an individual animal. The cyan solid lines indicate the second-order of linear model (A) and mixed-effects model (B) and shaded regions are of 95% CIs. Lower panel: spike trains of the representative neurons (located at dark arrows in the upper panel of Fig 4B) which show periodic and synchronous firings for either low or high couplings but exhibits chaotic firings for intermediate couplings.

complex spike data and to investigate the effects of changes in this coupling on the dynamics of olivo-cerebellar activity.

To estimate the effective coupling among IO neurons, we determined the $g_i$ and $g_c$ values that allowed our model to best match the experimental complex spike data in conditions of normal (CON), high (PIX), and low (CBX) synchrony levels, which are assumed to correspond to normal, high, and low coupling among IO neurons. These values were then used to calculate the effective coupling and coupling coefficients between IO neurons under these conditions. In addition, we adopted a Bayesian model-averaging approach to examine the effect of synaptic input strength on the effective coupling. As the result, we obtained an estimate of the physiological range over which the coupling between IO neurons may vary.

The validity of our methods to estimate coupling strengths is supported by several observations. First, the direction of the changes in $g_c$ and $g_i$ between the control and each drug condition determined by the model matched the known effects of the drugs. Specifically, PIX is a GABAa receptor antagonist and the model best simulated the complex spike patterns in PIX by decreasing $g_i$. In contrast, CBX spiking patterns were best reproduced when $g_c$ and $g_i$ were both reduced. These changes match the pharmacology of CBX, which is generally known as a

gap junction blocker, but also blocks GABAa receptors [59]. Second, reflecting these changes $g_c$ and $g_i$, $g_{eff}$ rose with PIX and fell with CBX, leading to increased and decreased coupling coefficients, respectively, which is again consistent with the observed experimental changes in complex spike synchrony caused by these drugs. Third, our coupling coefficient results agree with those of an *in vitro* slice study in which the effect of GABA on IO coupling coefficients was directly measured [49]. Specifically, baseline coupling coefficients in IO slices (CC = 0.021 ± 0.02, cf. S1 Table in [49]) closely match the values we obtained for the PIX condition (CC = 0.019 ± 0.006, Fig 1D), which is consistent with the fact that in both cases there is little to no GABAergic activity (the lack of spontaneous GABAergic activity in the slice was confirmed by the lack of effect of applying gabazine, a GABAa antagonist, on slice activity). Moreover, when GABAergic fibers were activated optogenetically in the study of [49] (CC = 0.012 ± 0.013), this created a situation analogous to the CON condition in our study (CC = 0.012 ± 0.003), because deep-cerebellar cells are spontaneously active in the anesthetized animal. In both studies, an approximate doubling of the coupling coefficients was found for the conditions where GABAergic activity was reduced or blocked. Thus our estimated coupling strengths values fall within the range of physiologically realistic values.

The validity of our approach also rests on the assumption that complex spike synchrony observed in the recordings is primarily due to the electrical coupling of IO neurons as opposed to some other source, such as correlated activity in afferents [72]. Indeed, in awake animals, synaptic input to the IO can limit the impact of gap junctional coupling within the IO [73]. Furthermore, the synchrony is reduced under isoflurane anesthesia compared to the awake state [20]. However, the evidence suggests that in our recordings, and probably under multiple physiological conditions, complex spike synchrony patterns largely reflect the effective coupling among IO neurons. We note that significant levels of complex spike synchrony remain after blocking GABAergic and/or glutamatergic afferents to the IO [33–35]. Moreover, complex spike synchrony depends on gap junction coupling, as it is lost or greatly decreased by pharmacological block of the gap junctions and is absent in connexin36 knockout mice in which IO neurons are not coupled [30,32,74]. Thus, synchrony in spontaneous complex spike activity requires electrical coupling of IO neurons and can occur in the absence of IO afferent activity. While these experiments were obtained in anesthetized animals, similar patterns of synchrony are found for spontaneous complex spike activity in awake animals [63]. Thus, the relationship between synchrony and electrical coupling seems to be broadly valid. Of course, complex spike synchrony could be driven by highly synchronized afferent activity in certain situations. In fact, complex spike activity driven by electrical stimulation of the motor cortex does show higher levels of synchrony. However, even in this case, the spatial distribution of synchrony still matches the spontaneous distribution set by electrical coupling [34]. In sum, the basic patterns of complex spike synchrony seem to strongly reflect the coupling pattern within the IO, even in the face of highly correlated afferent activity.

## Synchrony as a mechanism for controlling dimensionality

Our analysis of complex-spike data shows that controlling the effective coupling between IO neurons may be a mechanism for controlling the dimensionality of olivo-cerebellar activity. In particular, we found that increased electrical coupling between IO neurons decreased the dimensionality of IO firing dynamics. Dimensionality reduction has long been considered one of the core computations in the brain [75–79]. Our study provides direct evidence that electrical coupling among neurons can control the dimensionality of the population activity by modulating the synchrony of the neural code. Quantitatively, the approximately two-fold reduction in dimensionality from the PIX to the CON condition was highly comparable to that of stimulus-evoked activity of cortical neurons under different stimulus conditions and in varied tasks

[76,79]. We note, however, that additional mechanisms could work in parallel to effectively control the dimensionality, such as pruning of irrelevant inputs [80]. In the olivo-cerebellar system, in particular, climbing fiber-Purkinje-cell synapses are gradually eliminated based on IO activity during development [81,82]. We further note that our proposal of dimension reduction of an oscillatory system via coupling-induced synchronization contrasts other neural networks (e.g., auto-encoders) in that we propose a framework of neural communication among neurons for transmitting information rather than approximating a function that maps the data from high-dimensional space to low-dimensional space (i.e, encoding/decoding). In most artificial neural networks, such encoding/decoding scheme creates a black box on their mechanisms in reducing the dimension in the data. For instance, there is no meaningful link between the weights and the function being approximated or which variables in the data are irrelevant is an open problem. In contrast, the core idea of our proposal is that coupling provides a biologically plausible mechanism to achieve stable and reliable transitions between different oscillatory regimes [83]. Such framework allows the olivo-cerebellar system to dynamically control of the dimensionality depending upon the required task.

### Intermediate coupling strengths induce chaotic firings in inferior olive neurons

Our results also show that intermediate ranges of electrical coupling maximize chaotic dynamics. The model suggests that low complexity entropies found at weak and strong coupling levels are due to two different mechanisms. In the first mechanism, strong couplings synchronize the neurons and thus reduce the entropy of the network. In the second mechanism, weak couplings enhance asynchrony in the network. However, in the limit of no coupling, IO neurons do not interact. In this case, the whole network possesses a quasi-periodic solution when natural frequencies of different IO neurons have irrational ratios. It is known that the maximum Lyapunov exponent of quasi-periodic solutions is zero [84]. Thus, no chaotic behavior is expected in the no coupling condition, and very weak coupling should lead to similar dynamics. In contrast to the scenario of strong and weak couplings, moderate interactions of the neurons via intermediate coupling strengths induce asynchronous and irregular spiking activity and thus maximize the entropy [85]. We note that, in addition to the coupling strengths, there exists several factors that may affect chaotic dynamics of IO neurons. We found, in simulations, that the landscape of the complexity entropy, with respect to inhibitory conductance $g_i$ and gap-junctional conductance $g_c$, changes as the synaptic input level varies (S10A Fig). However, an inverted-U curve of complexity entropy as a function of effective coupling was observed in all the synaptic input levels tested, indicating the robustness of our finding that intermediate coupling strengths induce chaos (S10B Fig).

The finding of an inverted U curve in both the model and experimental data are consistent with the "chaotic resonance" hypothesis, according to which chaotic firing increases information transmission despite the low firing rates of IO neurons [43]. We have previously proposed, and shown in simulations, that such chaotic firing may be useful to enhance cerebellar learning by increasing the error transmission capability of the olivocerebellar system [45]. In agreement with this view, a previous combined *in vitro* and *in vivo* study of balanced excitatory/inhibitory cortical networks showed that the entropy of neural activity and mutual information between stimulus and response are maximized [86].

### Modulation of electrical coupling is a key parameter of olivocerebellar activity

Our results support the view that the efficacy of coupling between IO neurons is regulated by synaptic inputs to the IO, particularly those that terminate on the gap junction-coupled spines

[25]. Both excitatory and inhibitory synapses are present and may provide complementary mechanisms for controlling the strength of coupling between IO cells [87]. In the first mechanism, presynaptic GABAergic terminals control the efficacy of electrical coupling [25,26,49,56,88]. Note that GABAergic sources may also have an inhibitory effect on IO activity in general, and on their subthreshold oscillations in particular [89]. Thus, the changes in complex spike activity probably reflect both changes in coupling strength and excitability of IO neurons. We examined that possibility by varying the levels of the inhibitory synaptic inputs in the model. The effective coupling strengths estimated from the IO spiking data was slightly varied across the individual models, indicating the possible effect of GABAergic sources on excitability of IO neurons. However, we also found consistent effects of the drugs on the coupling strengths (i.e. increased by PIX and decreased by CBX), suggesting that the effect on complex spike activity reflects the drugs' effects on effective coupling strength directly (S4C Fig). In the second mechanism, glutamatergic synapses may control the strength of coupling between IO cells through both AMPA and NMDA-receptor mediated actions. It has been shown that blocking of AMPA receptors restricts and modifies the distribution of complex spike synchrony [34,35] and that activation of NMDA receptor strengthens coupling between weakly-coupled IO neurons thereby expanding the coupled IO network [36,37]. Thus, both of these mechanisms suggest that the olivo-cerebellar system can dynamically control the synchrony level of their corresponding climbing fiber inputs through regulating the coupling strength between IO neurons.

## Changes in dimensionality enable changes in modes of motor learning and control

The view that olivo-cerebellar axons carry error-based signals that induce plasticity at the parallel-fiber Purkinje cell synapses has received extensive experimental support since it was proposed on theoretical grounds by Marr and Albus (see Introduction). In addition, animal and human experimental data support the importance of electrical coupling for cerebellar learning [90,91]. Our results are compatible with the view that by modulating electrical coupling of IO neurons the dimensionality of olivo-cerebellar activity is adaptively optimized for different modes of motor learning. While plasticity of parallel fiber synapses with any particular PC may not necessarily depend on the dimensionality of IO activity, dimensionality and synchrony may have an influence on the distribution of plasticity across PCs, and this may be a key parameter of motor learning. For example, in the early phase of learning, motor commands are far from the desired ones. As a result, the Purkinje cells would be strongly modulated by large sensory inputs due to large error signals and their activity would inhibit cerebellar nuclei-IO neurons, thereby decreasing GABA release in the IO. Thus, IO neurons would be strongly coupled, and the dimensionality of olivo-cerebellar activity would be low. Because of this low dimensionality, the IO network would respond only to low-frequency components of the error signals, which would convey only the gross features of the motor commands. However, the strong coupling would allow widespread synchrony among IO neurons and potentially lead to parallel-fiber-Purkinje-cell synaptic plasticity (LTP or LTD) among large numbers of PCs simultaneously, resulting in fast but coarse learning. In contrast, in the late phase of learning, as the motor error becomes smaller, error-driven Purkinje cell activity would decrease, allowing increased activity of cerebellar nucleo-olivary neurons. This would result, in turn, in reduced IO coupling, lower synchrony, and thereby higher dimensionality. Furthermore, the decreased synchrony potentially would allow the occurrence of chaotic resonance to enhance information transmission of the error signals [41–45], which would overcome the constraint of low IO firing rates [55,92]. De-synchronized, high-

dimensional IO activity may optimally select the sites at which LTD/LTP occurs, and thereby allow more sophisticated learning, resulting in fine tuning of motor commands [45,46,93].

In addition to its effect on plasticity and learning, it is thought that the IO activity contributes directly to ongoing motor outputs throughout the learning process because changes in synchrony levels affect cerebellar nuclear cell activity directly [21,61,94]. Early on, highly synchronous activity, perhaps triggered by error signals, would trigger relatively crude corrective movements. Later on, more restricted synchrony patterns would convey high dimensional signals to be used for the fine grain motor control commands that are needed for precise motor coordination [19]. Note that the motor learning and control functions of the IO are not mutually exclusive, and the current consensus is that the olivo-cerebellar system contributes to both functions. Indeed, conceptual proposals have been made to integrate these two functional roles [39,95,96]. Additional modeling studies, in which IO and cerebellar networks are embedded into realistic motor systems with multiple muscles and joints will be needed to fully integrate and understand the role of the IO with its varying coupling strengths into motor learning and control.

## Methods

The recording experiments were performed in accordance with the National Institute of Health Guide for the Care and Use of Laboratory Animals. Experimental protocols were approved by the Institutional Animal Care and Use Committee of New York University School of Medicine.

### Experimental data

The analyses were performed on a subset of data obtained in two prior series of experiments in ketamine/xylazine anesthetized female, Sprague-Dawley rats that involved either injection of picrotoxin (PIX) or carbenoxolone (CBX) to the IO to block GABA-A receptors or gap junctions, respectively [30,33,34]. The specific experiments were chosen primarily on the basis of having typical complex spike activity in control and a large change in activity in response to the drug injection.

Details of the experimental procedures can be found in the original reports. In brief, a rectangular array of glass microelectrodes was implanted into the apical surface of crus 2a. The arrays typically contained 3–4 mediolaterally running rows and up to 10 rostro-caudally running columns, with an interelectrode spacing of ~250 μm. Electrodes were implanted to a depth of ~100 μm below the brain surface such that complex spikes from individual Purkinje cells were recorded. In each experiment, spontaneous complex spike activity was recorded during an initial control period. Following the control (CON) period, the IO was located by lowering a microelectrode through the brainstem under stereotaxic guidance until activity characteristic of IO neurons was observed. The microelectrode was then replaced by an injection pipette containing the drug solution that was lowered to the same location as the site where IO activity was found. A slow injection of drug solution was then performed (~1 μl over 5–10 min). The drug conditions analyzed were recorded after completion of the injection and a clear change in activity was observed. The multielectrode arrays recorded from 10–30 Purkinje cells in each of the CBX experiments (n = 6 animals), and from 16–42 Purkinje cells in the PIX experiments (n = 3 animals).

The effect of CBX and PIX on complex spike activity often varied among cells within an experiment. This was likely due to the Purkinje cells in different parts of the array receiving climbing fibers from different regions of the IO, that the drugs were injected at a single point within the IO, and that drug concentration (and therefore efficacy) will fall with distance from

the injection site. Indeed, the IO is an extended structure (particularly in the rostrocaudal axis where it is ~2 mm long). We therefore considered the effects of the drugs when selecting the neurons for analysis. That is, Purkinje cells that exhibited significant changes in complex spike firing rate, measured as the mean number of spikes per second, between the control and drug conditions were selected. For CBX, the criterion was a 50% decrease and for PIX it was a 50% increase (S2 Fig). In total, we analyzed spike train data from 500-second long periods for the control and drug conditions for each neuron (neurons/condition: control, n = 100; PIX, n = 47; CBX, n = 53).

## IO network model

The IO neuron model is a conductance-based model [52] extended via addition of glomerular compartments comprising electrically coupled spines [56]. The network model consisted of an array of 3x3 IO neurons, each of which was mutually connected to its four neighboring neurons by a gap junction from one of its spines to one of its neighbor's represented by the gap-junctional conductance $g_c$, whose strength was drawn from a uniform distribution with the maximum deviation set at ± 20% of the mean (S3A Fig).

We simulated spike data of the nine cells with stepwise changes of two model parameters: inhibitory synaptic conductance $g_i$, and coupling conductance $g_c$. These two parameters were both varied in the range of 0–2.0 mS/cm$^2$ with an increment of 0.05 mS/cm$^2$. We generated a total of 41x41 = 1681 sets of 500-second long simulated spike trains. The simulated spike data for each variation of $g_i$ and $g_c$ was then compared with the actual spike data, and the parameters whose firing dynamics best fit to that of individual neurons in the control, PIX, and CBX conditions were selected as the estimated values (see below for details). Because the effect of the axial conductance of the spines, $g_s$, is equivalent to that of the gap-junctional conductance, $g_c$, in determining the amount of current will flow across the gap junction, $g_s$ does not need to be estimated from the data and thus was fixed at 0.1 mS/cm$^2$ [56]. To better account for excitability of the neurons *in vivo*, the inward sodium current conductance $g_{Na}$ was set as 110 mS/cm$^2$, which has been shown to induce robust chaos in the model [43]. All of the soma, dendrite, and spine compartments respectively receive 10, 80, and 10 excitatory and inhibitory synapses driven by Poisson spike generators [56]. The numbers of synapses are roughly proportional to the surface areas of the three compartments.

## The segmental Bayes inference for estimating the effective coupling from a single model

Under simplified assumptions, the effective coupling, $g_{eff}$, between two IO neurons was calculated from the axial conductance of the spines $g_s$, inhibitory conductance $g_i$ and gap-junctional conductance $g_c$ as in [57]:

$$g_{eff} = \frac{g_s}{2g_c + g_i + g_s} g_c. \tag{1}$$

This equation implies that to estimate the effective coupling $g_{eff}$, we need to estimate both the coupling conductance $g_c$ and the GABA conductance $g_i$ reliably for each of the three datasets CBX, CON, and PIX. For that purpose, we previously developed a Bayesian method that contains two steps [51]. In the first step, the parameters are estimated for each 50-second time-segment of individual neurons, allowing the parameter values to vary in time. This compensates for inevitable mismatch in the firing patterns between the model and the data. In the second step, a single set of parameter values is estimated for the entire set of time-segments of

individual neurons by a hierarchical Bayes framework. Below, we outline the segmental Bayes method (for a detailed description, see [51]).

First, the firing dynamics of the spike data were characterized by a feature vector composed of a total of sixty-seven spatiotemporal features, e.g., firing rate, local variation [58], cross-correlation, auto-correlation, and minimal distance [97]. Principal component analysis (PCA) was then conducted to remove the redundancy of those features. The Bayesian inference aims to inversely estimate the conductance values from the top three principal components, which accounted for 55% of the data variance (S3B Fig). To compensate for the modeling errors, i.e. differences in the complexity of firing patterns between the model and actual neurons, we divided the spike data of each neuron into short time-segments under the assumption that segmental estimates of individual neurons fluctuated around a single neuronal estimate with a normal (Gaussian) distribution. The conductance values of individual neurons can be estimated by a hierarchical Bayesian framework (S3C Fig). Here, the segment size, 50 seconds, was optimized so that the variance of firing frequency across segments was minimal [56]. We have shown that the segmental Bayes algorithm minimizes the fitting between experimental and simulated spike data [51], and further confirmed, by simulations, that it indeed minimizes the estimation errors compared to other conventional methods–including the non-segmental Bayes inference, which finds the estimates once across the entire spike data, and the minimum-error algorithm, which directly finds the closest match in the feature space [98].

## Model-averaging estimation of the effective coupling between IO neurons

We found that the firing frequency of inhibitory synaptic noise inputs significantly affects the spiking behavior of the IO model and thus the estimation results. To reduce the uncertainty in estimates of $g_c$ and $g_i$, we therefore adopted the segmental Bayes algorithm by using a model-averaging approach as follows (for review, see [99]). We first simulated four models with the firing frequency of inhibitory synaptic inputs of 10, 20, 50 and 70 Hz, which are observed in slices of cerebellar nucleo-olivary neurons [100]. Next, we conducted the segmental Bayes to estimate posterior probability of $g_i$ and $g_c$ for each model.

$$P(g|y, m_i) \propto P(y|g, m_i)P(g|m_i), \qquad (2)$$

where $P(g \mid y, m_i)$ is the posterior probability of the conductance $g = (g_i, g_c)$, $y$ is the feature vectors extracted from the spike data, and $m_i$ is the $i$th selected model ($i = 1\ldots4$). We then mixed the posterior probabilities with the weights proportional to the model evidence as follows:

$$
\begin{aligned}
P(g|y) &= \sum_{i=1:4} P(g|y, m_i)P(m_i|y), \\
P(m_i|y) &\propto P(y|m_i)P(m_i), \\
P(y|m_i) &= \int_g P(y|g, m_i)P(g|m_i)dg, \\
P(m_i) &\propto 1,
\end{aligned}
\qquad (3)
$$

where $P(g \mid y)$ is the mixed probability for an individual neuron and $P(y \mid m_i)$ is the evidence of the $i$th model. Here, all models are treated equally with a non-preference prior $P(m_i)$. Finally, the point estimates of $g_i$ and $g_c$ were computed by marginalizing the mixed posterior probabilities (S4B Fig).

## Calculation of the synchrony for individual neurons

The spike train of a neuron was binned into $X(i)$, where $i$ represents the time step ($i = 1,\ldots,T$), with $X(i) = 1$ if the spike occurs in the $i$th time bin; otherwise, $X(i) = 0$. The synchrony of two

different neurons, $x$ and $y$, was calculated as the cross-correlation coefficient at zero-time lag:

$$C_{x,y} = \frac{\sum_{i=1}^{T} \bar{X}(i) \bar{Y}(i)}{\sqrt{\sum_{i=1}^{T} \bar{X}(i)^2 \sum_{i=1}^{T} \bar{Y}(i)^2}},$$ (4)

$$\bar{X}(i) = X(i) - \frac{1}{T}\sum_{j=1}^{T} X(j), \quad \bar{Y}(i) = Y(i) - \frac{1}{T}\sum_{j=1}^{T} Y(j),$$

where $\bar{X}(i)$ and $\bar{Y}(i)$ are the normalized forms of $X(i)$ and $Y(i)$ to account for the firing frequency. Here, the two spikes were considered synchronous if their onsets occur in the same 10 milli-second bin. The synchrony level of an individual neuron $x$ was computed as the mean of $C_{x,y}$ for all neurons $y \neq x$ in the same animal.

## Estimation of the coupling coefficient by simulations

To examine whether the estimates of effective coupling strengths were biologically realistic, we computed the coupling coefficients (CCs) for the model neurons as follows. After hyperpolarizing all neurons to -69 mV by injection of $I_{hyp}$ = -1 μA/cm$^2$ to increase responsiveness, we injected a step current $I_{cmd}$ = -1 μA/cm$^2$ in the soma of the center neuron. We computed the CCs as the average ratio of change in steady state membrane potentials of this "master" cell and its four neighboring cells (S5A Fig). We generated and computed the CCs for hundreds of $g_i$ and $g_c$ values over the range that the estimated conductance of the data was distributed and found a strong positive correlation between the effective coupling and the CC (S5B Fig, $R^2$ = 0.8, p < 0.0001). CC was determined by transforming the $g_{eff}$ value with the fitted model of $g_{eff}$ vs. CC (S5B Fig).

## Estimation of the dimensionality of neural firings

The dimensionality can be considered as the minimal dimensions necessary to provide an accurate description of the neural dynamics. Principal component analysis (PCA) has become the most widely used approach for determining this, because it enables neural dynamics to be represented in a lower dimensional space [79]. Here, we adopted this approach for estimating the dimensionality of the firing activity of a small number of neurons, like the numbers in the recording arrays.

We first sampled 50-s long spike trains using sampling intervals of 10 milli-seconds, from which the firing rate vectors of all neurons were computed (S6A Fig). Firing rate vector in each sampled window corresponds to an observation in the $N$-dimensional space, where $N$ is the number of ensemble neurons. Then, PCA was applied to estimate the dimensionality as [60]:

$$d = \frac{1}{\sum_{i=1}^{N} \tilde{\lambda}_i^2},$$ (5)

where $\tilde{\lambda}_i = \lambda_i / (\sum_j \lambda_j)$ are the principal eigenvalues expressed as the amount of variance explained (S6A Fig), and $\lambda_i$ is the $i$th eigenvalue of the covariance matrix of the firing rate vectors.

To test whether dimensionality was sensitive to the duration of the sampled window, windows with duration of 10–50 seconds were analyzed. No significant different values were found (S6B Fig), probably because IO firing rates are stable across each condition. However, it has been shown that dimensionality estimation depends on the number of ensemble neurons $N$. Specifically, $d$ is underestimated for small $N$ but becomes independent of $N$ for sufficiently large $N$ [79]. After data selection (see above), the number of IO neurons in each animal is

N = 4–20, which is likely to suffer from the under-sampling bias. Thus, to compare dimensionality among the animals, we normalized it by the number of selected IO neurons in individual animals (i.e normalized $d = d/N$).

## Computation of the complexity entropy

The Lyapunov exponents quantify the sensitivity of a dynamical system to initial conditions, and thus are often used as indicators of chaos [65,66]. A number of methods have been developed to compute the Lyapunov exponents from time series with a fixed sampling interval [67,68]. Those methods, however, are not applicable for our IO data because computation of Lyapunov exponents requires access to continuous variables, which is not the case in our discrete IO spike sets. We therefore adopted a previously proposed approach [70] that approximates the Lyapunov exponents via a recurrence plot by using the edit distance of spike trains [101]. Our method requires computing the modified edit distance of the spike trains [97] and its recurrence plot [102,103]. The edit distance of two derived windows is defined by a total minimal cost for converting one window to the other [101]. Allowed operations include deletion or insertion of events (both cost 1 for each event), and shift of events (cost 20% the amount of shifting in second for each event). The complexity entropy [69] was computed from the distribution of the length of diagonal lines in the recurrence plot (see S8 Fig for illustration of the complexity method).

Specifically, we first sampled the spikes trains in windows of 50 seconds and computed the edit distance for all pairs of sampled windows. To resolve the issue of discontinuity induced by the difference in the number of spikes in two sampled windows, we adopted a modified version of edit distance computation as in [97]. Briefly, for each sampled window, we took into account the spikes that occur immediately before and/or after the time window, thus resulting in four derived windows. We then computed the edit distance for a total of 16 (4x4) derived pairs of the two sampled windows and temporarily assigned the minimum value as edit distance between them. The edit distance for all pairs of sampled windows of 50 seconds with an interval of 2 seconds constitutes a two-dimensional distance matrix. We then updated the edit distance matrix by the shortest distance connecting any two sampled windows–S8A Fig. The recurrence plot is constructed by binarizing the edit distance matrix, with the distance values smaller than a predefined threshold as 1, and the others else as 0 [102]. The threshold was determined so that 10% of data points in the distance matrix were 1, as in [103]. Next, we extracted the frequency distribution of the length of the points 1 that form diagonal lines in the recurrence plot. The Shannon entropy of that distribution has been shown to be inversely proportional to the largest Lyapunov exponent [69]. We thus used the inverse of Shannon entropy as a measure of chaos for the spike data.

To validate that complexity entropy is an indicator of chaos, we generated noise-free simulation data and computed the correlations between complexity entropy and the Lyapunov indexes (S8B and S8C Fig). Note that this approach is possible for the simulation data because we have access to the continuous trace of the membrane potential. Specifically, we first removed the noise in the synaptic inputs, and simulated 500-second spike trains for more than 100 conductance values ($g_i$ varied in 0–1.0 mS/cm$^2$ and $g_c$ in 0–2.0 mS/cm$^2$) and estimated the complexity entropy from the simulated spike trains. Next we computed the Lyapunov exponents of the IO model by the method of [104], and then extracted the largest component, $\lambda_1$, as well as the Lyapunov dimension, $D_{KY}$, as these are two direct indicators of chaos [105].

## Statistical analysis

Unless specifically stated elsewhere, all data is reported as *mean ± std*. The non-parametric Kruskal-Wallis one-way analysis of variance was used to test whether data groups of different sizes originate from the same distribution.

## Code availability

The simulation code of the IO network model is hosted publicly on github, accessible via https://github.com/hoang-atr/io_model. The MATLAB implementations of the segmental Bayes algorithm [51] and the complexity entropy method [70] are available upon request from the corresponding authors.

## Supporting information

**S1 Fig. Inferior olive firing data set for all animals.** A: Spike data in 50 second of 10 representative neurons in 9 animals with the physiological conditions (CBX and PIX) in the right and the control condition (CON) in the left columns.
(TIF)

**S2 Fig. Data selection by changes in the firing rate.** A: the histograms of firing rate change by drug treatments (decreased by CBX and increased by PIX) compared to the CON condition. The red lines indicate the thresholds (50%) for selecting the neurons for analysis. B: pseudo-color maps show the firing rate change by drug treatments of the neurons in the micro-electrode arrays for six CBX animals (top two rows) and three PIX animals (bottom row). Red asterisks indicated the selected neurons, whose firing rate changes exceed the thresholds.
(TIF)

**S3 Fig. Estimation of the conductances from spiking data using IO model and segmental Bayes algorithm.** A: left, electrical circuit equivalents of the soma (S), dendrite (D) and spine compartments (SP) of a model IO neuron. middle, the connection of two IO neurons via a gap junctional conductance $g_c$ that connects the spine compartments. right, The IO network, which consists of 3x3 neurons, each of which is connected to its four neighboring neurons as shown. B: Left, five major features (FR = firing rate, ACG1 = auto-correlogram in 50 ms bin, CCG1 = cross-correlogram in 50 ms bin, MD1 = the first fraction of the minimal distance distribution, LV = local variation) extracted from spiking data of the three conditions (see [56] for detailed definitions of the features). Each feature was normalized by the mean value of the CON level. Right, the top two principal components of the extracted features. C: Flow chart of conductance estimation for each neuron. To account for highly non-stationary of the spike patterns in the three data condition, we divided the spike data of each neuron into small time-segments, applied the Bayesian inference to estimate $g_i$ and $g_c$ for every segment under the assumption that segmental estimates were drawn from a single neuronal estimate of a normal (Gaussian) distribution with unknown mean and the variance as a prior (left). The variance was optimized so as to maximize the fit of the data and the model in the PCA space. The posterior estimation for a representative neuron's $g_i$ and $g_c$ conductances (right). A broad probability distribution was found when the variance was relaxed but a much smaller distribution resulted when optimized variance was used [51].
(TIF)

**S4 Fig. Improving the parameter estimates via Bayesian model-averaging.** A: PCA error rates of the $g_i$ and $g_c$ estimates by the segmental Bayesian inference averaged for the entire IO neurons for CBX, CON, and PIX conditions for four different models (color bars) in comparison with the previous model (black bar, [56]). The error bars are of 95% CIs. B: Posterior probabilities of a representative IO neuron by individual models and the mixed posterior probability with the weights determined by the evidence of Bayesian inference. C: boxplots show estimates of the effective coupling $g_{eff}$ of the three data conditions in the four individual models.
(TIF)

**S5 Fig. Estimation of the coupling coefficient (CC) by simulation.** A: We injected a current pulse of -1 μA/cm$^2$ to a cell and recorded the steady-state voltage change of this "master" cell and its four post-junctional cells. B: We computed the CCs for hundreds of $g_i$ and $g_c$ values in the range over which the estimated conductances of the data distributed, and found a strong positive correlation between the effective coupling and the CC ($R^2 = 0.8$, $p < 0.0001$). Note that the non-linear fit represents the nature of deriving $g_{eff}$ from $g_i$ and $g_c$ following Eq (1). (TIF)

**S6 Fig. Dimensionality estimation for the spike data of ensemble neurons.** A: Illustration of the principal component analysis (PCA) for the firing rate vectors extracted from 50-second windows of three neurons of Animal #6 in the CON condition. The estimated dimensionality $d = 1.86$ (dashed dark line, Eq 5), indicates that the approximately 2-dimensional subspace (shaded gray plane) can explain more than 90% of the variance of neural firing dynamics. B: Estimating dimensionality (Eq 5) with varied window lengths from 10–50 seconds for 9 animals in the three data conditions showing the robustness of dimensionality estimation against the window length. The error bars are of 95% CIs. (TIF)

**S7 Fig. Change in synchrony is linearly coupled with change in dimensionality.** We first divided the complex spike data of 9 animals in the control condition into a series of shorter segments using a moving window of length 5, 10, 20, or 50 s, whose start incremented in 1 second steps. Next, we computed the dimensionality and the synchrony (both in 10 milli-second timescale) for each segment (A) and measured the correlation coefficient between those two metrics (B). The correlation coefficients were negative for almost all animals and window lengths (C). These results provide clear evidence that variations of synchrony within the physiological range are negatively correlated with changes in dimensionality. Note that the control data of spontaneous complex spike activity in our study is within the range of physiologically occurring synchrony levels. A: time-varying synchrony (left ordinate) and dimensionality (right ordinate) in successive, overlapped 10-s sliding windows of the control recording from animal #3. B: Scatterplot of correlation versus dimensionality values shown in A shows the presence of a significant negative correlation ($r = -0.6$). C: synchrony vs. dimensionality correlation coefficients for 9 control animals and all tested window lengths. (TIF)

**S8 Fig. Computation and validation of the complexity entropy method.** A: Illustration of edit distance computation between two sampled spike windows shows a sequence of elementary steps that convert the spike window (a) into (b). Each bar represents one spike. Allowed operations include deletion of a spike (shown in red), insertion of a spike (shown in green), or shifting a spike in time (blue arrows). Computation of edit distance for continuous sampling windows for the entire spike train constitutes the edit distance matrix. Then, the recurrent plot is constructed by binarizing the edit distance matrix. The points, whose values are smaller than the threshold, were plotted as white dots, otherwise as black dots. Complexity entropy is computed as the inverse of Shannon entropy, in terms of frequency distribution of the length of the diagonal lines of white dots [69]. B–C: Complexity entropy measured for a total of a hundred of parameter values (black crosses) in noise-free simulations showed strong positive correlations with the largest Lyapunov exponent $\lambda_1$ (regression model: $\lambda_1 \sim 1 + entropy$, $R^2 = 0.4$, F-test: $p < 0.0001$, S8B Fig) and the Lyapunov dimension $D_{KY}$ ($D_{KY} \sim 1 + entropy$, $R^2 = 0.48$, F-test: $p < 0.0001$, S8C Fig). Solid cyan lines represent the fit of linear models with 95% CIs (dashed cyan lines). (TIF)

**S9 Fig. Validation of the inverted U-shaped curves.** We investigated whether intermediate couplings maximize the complexity entropy by applying a non-parametric Gaussian Process regression model, which does not assume an explicit relationship between the coupling and the complexity entropy. Still, we observed inverted U-shaped curves maximized at around $g_{eff}$ = 0.033 mS/cm$^2$ for both the model (A) and the data (B). In sum, these results support the inverted U-shaped relationship between the effective coupling and complexity entropy. The right ordinates of A–B represent the first Lyapunov exponents approximated from the simulation data (see S8B Fig), indicating that intermediate couplings induce chaos. The shaded regions are of ±sem.
(TIF)

**S10 Fig. Effect of inhibitory synaptic input on complexity entropy.** A: the pseudo-color heatmap of the complexity entropies, averaged across 9 model neurons, for all pairs of ($g_i$, $g_c$) in the range of 0–2 mS/cm$^2$. The frequency of inhibitory synaptic input noise was varied in the range of 10–70 Hz. B: complexity entropy vs. effective coupling $g_{eff}$. In each model, a second-order regression ($entropy \sim 1 + g_{eff} + g_{eff}^2$) fit was shown by black thick line. The second-order coefficient was negative and significant for all the models indicating the robustness of the inverted-U curve.
(TIF)

## Author Contributions

**Conceptualization:** Mitsuo Kawato, Nicolas Schweighofer.

**Data curation:** Eric J. Lang.

**Investigation:** Huu Hoang, Eric J. Lang, Mitsuo Kawato, Nicolas Schweighofer.

**Methodology:** Huu Hoang, Yoshito Hirata, Isao T. Tokuda, Kazuyuki Aihara, Keisuke Toyama, Mitsuo Kawato.

**Software:** Huu Hoang, Yoshito Hirata.

**Supervision:** Mitsuo Kawato, Nicolas Schweighofer.

**Visualization:** Huu Hoang.

**Writing – original draft:** Huu Hoang, Eric J. Lang, Yoshito Hirata, Kazuyuki Aihara, Mitsuo Kawato, Nicolas Schweighofer.

**Writing – review & editing:** Huu Hoang, Eric J. Lang, Mitsuo Kawato, Nicolas Schweighofer.

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
