## [Decision Letter · Decision Letter 0]

3 Jan 2020

Dear Dr Hoang,

Thank you very much for submitting your manuscript 'Electrical coupling controls dimensionality and chaotic firing of inferior olive neurons' for review by PLOS Computational Biology. Your manuscript has been fully evaluated by the PLOS Computational Biology editorial team and in this case also by independent peer reviewers. The reviewers appreciated the attention to an important problem, but raised some substantial concerns about the manuscript as it currently stands. While your manuscript cannot be accepted in its present form, we are willing to consider a revised version in which the issues raised by the reviewers have been adequately addressed. We cannot, of course, promise publication at that time.

Please make sure to address the concerns of all three reviewers. In addition, I would like you to add at least in the discussion your view on the impact of dimensionality in olivary activity on the induction of LTD (by increase in CF activity) as well as that of LTP (by a decrease in CF activity) in the context of learning (see also the comments from reviewer 3).

Sincerely,

Chris De Zeeuw

Guest Editor

PLOS Computational Biology

Daniele Marinazzo

Deputy Editor

PLOS Computational Biology

[LINK]

Reviewer's Responses to Questions

**Comments to the Authors:**

Reviewer #1: It is an interesting study that provides new insights about how olivo-cerebellar loop processes information and how the timing and learning hypothesis of cerebellar function can be reconciled. However, I have some points that I would like you to address:

1) During the whole study (Introduction, results and discussion). It was not mentioned at all the contribution of excitatory synapses to the modulation of coupling strength in the IO (Turececk et al 2014, Mathy et al 2014). For example, I saw you mentioned the study of Turececk et al, 2014. However, you seem to bypass the main findings of that paper which is that the activation of NMDA receptors enhance electrical coupling in weakly coupled neurons in the IO. Taking those results into account, you could also say that the observed enhancement of electrical coupling during the application of PIX in rats is due to an enhancement of excitation rather than absence of inhibition. I think both introduction and discussion sections should be rephrased in such a way that also should be considered the possibility of the effect of excitatory synapses on coupling strength in the IO. This is very crucial to keep in mind because it is known the close proximity of excitatory and inhibitory terminals to the electrically coupled dendritic spines in the glomeruli (De Zeeuw, 1990).

2) Figure 1, panel A: Do you think that the possible significant difference found between CBX and control conditions can be due to the possible direct actions of carbenoxolone on GABAa receptors (Tovar et al, 2009. Journal of Neurophysiology).

3) Figure 1, panel B: How can you explain the non-significant difference found in the gap junctional conductance between the control and PIX group?

Reviewer #2: The inferior olive is crucial for setting the activity level of the cerebellum, affecting (motor) execution as well as (motor) learning. The authors hypothesize that adjusting the coupling strength between inferior olivary neurons changes the dimensionality of inferior olivary output. The authors integrate (previous) experimental data with computational modeling to demonstrate that their ideas are biologically plausible.

Compared to other brain regions, neurons of the inferior olive (IO) have extraordinarily strong gap junctional coupling. This strong coupling has long been considered to be the basis of synchronization of IO output, and thus of complex spike firing by the postsynaptic Purkinje cells. However, it is now clear that the impact of coupling depends on the behavioral state of the animal and is strongly affected by anesthesia (see e.g. Mukamel et al., Neuron 2009). In awake animals, synaptic input to the IO strongly limits the impact of gap junctional coupling within the IO (see e.g. Negrello et al., PLoS Comp Biol 2019). Indeed, the authors noticed this impact of synaptic input (lines 159-160), but largely ignored this observation during their study. While their idea – controlling the dimensionality by adjusting coupling strength within the IO – is interesting and appealing, the use of experimental data from anesthetized animals restricts the interpretational value of this study. Several studies, using predominantly optical recording techniques, but there are some electrophysiological studies as well, have published complex spike recordings of multiple Purkinje cells in awake animals and related to specific tasks. Relating the ideas of the current manuscript to (at least) one of the awake datasets would strengthen the study.

In Fig. 3, the authors suggest – but do not show directly – that synchrony is linearly related to dimensionality. This seems an intuitive, if not trivial, result. It would help to show whether changes in synchrony that are physiologically relevant are indeed coupled to changes in dimensionality. This would demonstrate whether IO synchrony is indeed modulated during different phases of motor adaptation and learning, as suggested by the last paragraph of the Discussion.

The authors present direct comparisons of biological data and modeling data (e.g., Fig. 2). This comparison would benefit from providing more details (also in the Figure; not only summarized in the text). While the impact of the drugs on firing rate is obvious, more relevant measures like synchrony are not illustrated in a direct comparison.

In short, the authors present an interesting hypothesis: that the degree of gap junctional coupling controls the dimensionality / entropy of the IO output via modulation of IO synchrony. Their data suggest that the amplitude of changes in gap junctional coupling required for having an impact on dimensionality / entropy are realistic given experimental data. However, the experimental data are recorded under anesthesia, which is known to affect IO synchrony, and variations in coupling are only linked to pharmacological intervention and not to behaviorally relevant changes, clearly limiting the relevance of this study in its current form.

Reviewer #3: I apologize in advance if the review sounds harsh.

The paper puts forward a story about the putative functional consequences of an adaptive modulation of gap junctional plasticity in the inferior olivein the course of learning. It extends a well-validated model of inferior olivary cells to a network in order to study the effect of gap junctional modulation on ‘dimensionality’ of the system.

I have multiple problems with the paper in its current form, and I am afraid I cannot recommend publication in PLoS Comp. Bio. The argumentation in the paper is rather shallow and repetitive, mostly recapitulating earlier arguments but with scant new evidence.

The whole introduction reads as a discussion of the functional consequences of an assumption. It heavily depends on a particular theoretical framework, that which upholsters the view that cerebellar function is to perform a basic form of supervised learning. It uses the word dimensionality profusely, but confusingly. It identifies reduction of dimensionality in the control problem with the reduction of dimensionality in synchronizing coupled oscillators, though the logics for this step are not spelled out. Furthermore, the authors argue here (and previously) that a reduction of dimensionality is a desirable end state for motor control, but that is not at all a given, and depends very much on the type of control problem (for example, smooth grasp requires higher correlation between output control variables).

Major points:

1. Heavy usage of abstract terms with vague and weak links to actual problems of motor control:

- The use of the word ‘dimension’ is too loose. There seems to be a confused use of the term ‘reduction of dimensionality’, where the authors seem to assume that the reduction of dimensionality of an oscillatory system due to synchrony is akin to what happens in neural networks (such as in auto-encoders). This identification is highly problematic, because it is not clear which are the variables being encoded/decoded.

- Substantial differences are expected in coupling conductances in real networks. Small perturbations would be sufficient to create mounting phase disparities in the cell populations, unless these perturbations are correlated.

- In an oscillatory system driven by noise, the use of chaos as a functional property is contentious. The time scales where sensitivity to initial conditions would have a measurable effect are long with respect to motor behavior. More plausible sources of phase variability exist which are ‘transient’, rather than asymptotic, as required by the use of the term ‘chaos’.

- Chaos in a slowly oscillating system would only be measurable in longer time scales (>>100ms) . But in long time scales the sensitive dependence of initial conditions will likely be swamped under the effects of (1) input noise (2) intrinsic plasticity (3) short term synaptic plasticity (4) changes of rectifying properties in gap junctions. In the absence of parameter space analysis

- Many processes influence and drive synchrony, most importantly, input correlations. The level of synchrony induced by coupling is likely smaller than the correlations from GABAergic and Glutamatergic sources.

- “Low dimension components of error signal” Usually the error signal is assumed to be a coarse representation of an organismic-level event.

- Similarly, it is not always clear what the DoF that the paper is referring to. Motor synergies highly constrain the number of DoF in motor output systems. Only twitches and the like fit into ‘reduction of DoF’.

2. Cerebellum as supervised learning machine: tacit acceptance of the theory that supervised learning in the cerebellum is driven primarily by the CF related LTD.

- The idea is that reduction of coupling conductance and synchrony co-occurs with training in the system presumes that the main learning mechanism in the cerebellum is CS driven PF plasticity, i.e., supervised learning. Though this is a common belief, there are continued and substantial problems with the idea. Particularly as one attempts to create more complex control systems with more degrees of freedom in 3D space, the requirements for a motor error signal rise substantially. Though this is not the place for a full discussion of the problems with this assumption, given the number of years that the hypothesis had to prove its ability to explain high level control and acquisition, a more careful presentation of the theory is called for.

- The paper uses the literature to justify the assumption that GABAergic sources reduce coupling between cells. In the paper mentioned, they have not controlled for the effects of inhibition in the measured reduction of coupling conductance. It remains highly plausible that reduction of coupling is simply a consequence of shunting of oscillations rather than a direct effect on the coupling values. Hence, adaptation of coupling is not induced by GABAergic sources, but simply a momentary reduction of oscillatory amplitude.

**Have all data underlying the figures and results presented in the manuscript been provided?**

Reviewer #1: Yes

Reviewer #2: No: At least not in the current submission.

Reviewer #3: Yes

PLOS authors have the option to publish the peer review history of their article (what does this mean?). If published, this will include your full peer review and any attached files.

Reviewer #1: No

Reviewer #2: No

Reviewer #3: No

---

## [Decision Letter · Decision Letter 1]

15 May 2020

Dear Dr. Hoang,

Thank you very much for submitting your manuscript "Electrical coupling controls dimensionality and chaotic firing of inferior olive neurons" for consideration at PLOS Computational Biology. As with all papers reviewed by the journal, your manuscript was reviewed by members of the editorial board and by several independent reviewers. The reviewers appreciated the attention to an important topic. Based on the reviews, we are likely to accept this manuscript for publication, providing that you modify the manuscript according to the review recommendations.

Sincerely,

Chris De Zeeuw

Guest Editor

PLOS Computational Biology

Daniele Marinazzo

Deputy Editor

PLOS Computational Biology

[LINK]

Reviewer's Responses to Questions

**Comments to the Authors:**

Reviewer #1: This new version of the manuscript has been improved substantially. You have answered all my questions and include my suggestions. Moreover, You have clarified some important concepts such as dimensionality which provides the reader with a better understating of the main message of this study.

I only have minor remarks that I would like you to address:

Legend of Figure 2: Should the CBX animal show ‘chaotic’ firing instead of ‘repetitive’ whereas the control animal should exhibit ‘repetitive’ firing and not ‘chaotic?.

Figure 3: Would you expect a different relationship between dimensionality and synchrony if you would extract the coefficient of variation of adjacent interspike intervals (CV2) instead of the average firing frequency in order to calculate the dimensionality of complex spike activity?.

Figure 3B: Can you explain why is there no significant difference between both the PIX and control group and CBX and control group?.

Discussion: The last part of the discussion is redundant when it’s mentioned the role of synchrony at early and late stages of learning. It can be more concise.

Reviewer #2: I would like to thank the Authors for taking the time to substantiate their manuscript.

Synchrony within the inferior olive, and thereby between complex spikes of downstream Purkinje cells, is an abundant phenomenon. In this paper, the authors argue that gap junctional coupling between IO cells can affect synchrony and that this in turn affects the dimensionality of the olivary-cerebellar system. Intuïtively, there has to be a relation between synchrony and dimensionality: all cells firing in perfect synchrony minimize and all cells firing chaotically maximize entropy. In the first round of review, I mainly referred to the biological relevance of the current study. In response to my question regarding the relation between synchrony and dimensionality (which was suggested, but not shown), the authors now demonstrate this relation in the new Fig. S7.

I am afraid that my concerns were not fully addressed. I remarked that such a relation between synchrony and dimensionality is (almost) trivial. I indicated that it would help to provide evidence that the synchrony varies under physiologically relevant situations to such a degree that changes in dimensionality become apparent. Related to this issue, I have also encouraged the inclusion of awake data.

In response to the latter remark (on the inclusion of awake data) the authors have expanded the Discussion, giving a better representation of the value of the anesthetized data used for this study. I would accept this solution. However, the question remains whether "IO synchrony is indeed modulated durig different phases of motor adaptation and learning" (see my remarks during the previous round). I do not feel that the new Fig. S7 helps to address this.

In other words, this study argues that IO dimensionality is related to IO synchrony (via gap junctions), and I agree with these conclusions, but leave me with the question to what extent this happens during real life. I would like to encourage the authors to address the expected variation in synchrony more explicitly.

**Have all data underlying the figures and results presented in the manuscript been provided?**

Reviewer #1: Yes

Reviewer #2: Yes

PLOS authors have the option to publish the peer review history of their article (what does this mean?). If published, this will include your full peer review and any attached files.

Reviewer #1: No

Reviewer #2: No
---

## [Editor Report · Decision Letter 2]

18 Jun 2020

Dear Dr. Hoang,

We are pleased to inform you that your manuscript 'Electrical coupling controls dimensionality and chaotic firing of inferior olive neurons' has been provisionally accepted for publication in PLOS Computational Biology.

Best regards,

Chris De Zeeuw

Guest Editor

PLOS Computational Biology

Daniele Marinazzo

Deputy Editor

PLOS Computational Biology

---

## [Editor Report · Acceptance letter]

22 Jul 2020

PCOMPBIOL-D-19-01931R2 

Electrical coupling controls dimensionality and chaotic firing of inferior olive neurons

Dear Dr Hoang,

I am pleased to inform you that your manuscript has been formally accepted for publication in PLOS Computational Biology. Your manuscript is now with our production department and you will be notified of the publication date in due course.

With kind regards,

Laura Mallard
